# Machine Learning of the Whole Genome Sequence of *Mycobacterium tuberculosis*: A Scoping PRISMA-Based Review

**DOI:** 10.3390/microorganisms11081872

**Published:** 2023-07-25

**Authors:** Ricardo Perea-Jacobo, Guillermo René Paredes-Gutiérrez, Miguel Ángel Guerrero-Chevannier, Dora-Luz Flores, Raquel Muñiz-Salazar

**Affiliations:** 1Facultad de Ingeniería Arquitectura y Diseño, Universidad Autónoma de Baja California, Campus Ensenada, Ensenada 22860, Mexico; perear@uabc.edu.mx (R.P.-J.); paredesg@uabc.edu.mx (G.R.P.-G.); miguel.angel.guerrero.chevannier@uabc.edu.mx (M.Á.G.-C.); 2Escuela de Ciencias de la Salud, Universidad Autónoma de Baja California, Campus Ensenada, Ensenada 22890, Mexico

**Keywords:** NGS, MDR-TB, SNPs, *rpoB*, *Mycobacterium tuberculosis*, ML, IA

## Abstract

Tuberculosis (TB) remains one of the most significant global health problems, posing a significant challenge to public health systems worldwide. However, diagnosing drug-resistant tuberculosis (DR-TB) has become increasingly challenging due to the rising number of multidrug-resistant (MDR-TB) cases, despite the development of new TB diagnostic tools. Even the World Health Organization-recommended methods such as Xpert MTB/XDR or Truenat are unable to detect all the *Mycobacterium tuberculosis* genome mutations associated with drug resistance. While Whole Genome Sequencing offers a more precise DR profile, the lack of user-friendly bioinformatics analysis applications hinders its widespread use. This review focuses on exploring various artificial intelligence models for predicting DR-TB profiles, analyzing relevant English-language articles using the PRISMA methodology through the Covidence platform. Our findings indicate that an Artificial Neural Network is the most commonly employed method, with non-statistical dimensionality reduction techniques preferred over traditional statistical approaches such as Principal Component Analysis or t-distributed Stochastic Neighbor Embedding.

## 1. Introduction

Tuberculosis (TB) is a treatable and preventable infectious disease caused by *Mycobacterium tuberculosis*. Despite its curability, this ancient illness remains a major global health concern due to its high incidence and mortality rates worldwide. As of 2021, an estimated 10.6 million people had developed TB, and 1.6 million had lost their lives to the disease. The regime of drug-susceptible and drug-resistant *M. tuberculosis* isolates demands a minimum of three to four antibiotics (rifampicin, isoniazid, ethambutol, and pyrazinamide) in combination, leading to complex patterns of drug susceptibility and resistance. The World Health Organization (WHO) estimates that globally in 2020, 71% of people diagnosed with bacteriologically confirmed pulmonary TB were tested for rifampicin (RIF) resistance, up from 61% in 2019 and 50% in 2018 [1]. In 2019, about 0.5 million DR-TB cases were reported worldwide, of which 78% were MDR-TB. MDR-TB is defined as resistant to at least RIF and isoniazid (INH), the most effective first-line antituberculosis drugs. It is estimated that one in four deaths caused by antimicrobial resistance is due to rifampicin-resistant TB. Treating drug-resistant TB is more complex than treating drug-susceptible TB.

A Drug Susceptibility Test (DST) is essential for proper antituberculosis treatment, avoiding complications, and significantly reducing the treatment period. Microbiological culture is the gold standard to evaluate the DR; however, it requires two to six weeks to obtain results and must be performed at a Biosafety Level 3. Consequently, most patients start antituberculosis treatment without DST information. To effectively treat DR-TB, a rapid and specific drug sensitivity test (DST) is necessary for selecting the appropriate TB treatment. This test helps identify the most effective treatment for the patient. [2]. Today, the World Health Organization (WHO) recommended molecular tests, such as Xpert^®^ MTB/RIF (Cepheid, Sunnyvale, CA, USA), Truenat^®^ MTB, and the MTB Plus system (Molbio Diagnostics, Goa, India), to use as initial tests for the diagnosis of TB and rifampicin-resistant TB. However, molecular DSTs cannot detect resistance profiles when mutations occur outside the target genetic region. On the other hand, Whole Genome Sequencing (WGS) is a technique that can compensate for this weakness. 

WGS allows the identification of the DR-TB profile-identified known mutations and can be used to propose new mutations that confer resistance when compared with a diverse amount of DST; it is accurate and provides a rich set of additional information for further analysis of new TB antibiotic development. Therefore, it is essential to explore the value of modern statistical approaches, such as Machine Learning (ML), which can analyze vast amounts of characteristics in large databases such as genomics and perform high-precision resistance classification. ML models can be utilized to analyze the whole genome sequencing of *M. tuberculosis* strains, helping predict resistance profiles and reducing the time delay in starting appropriate treatment. The increased availability of new artificial intelligence technologies, in particular ML and Deep Learning (DL), allows an approach to complex clinical databases, radiological images, and whole genomes to perform rapid detection and classification of the disease, support clinical decision-making, and contribute to quick and timely diagnosis. However, it is still being determined what model is recommended for these biological data or if the metrics are reported similarly and consistently. Furthermore, the several ways of grouping the resistance analyses by drug or treatment regimen make it challenging to compare them. There is no standardized method for analyzing sequence data to ensure a good result for resistance prediction. These methods can include, for example, analyzing mutations already known to confer drug resistance, analyzing the entire genome (considering only mutations or the whole genome compared against a reference), and analyzing specific genes. To better understand the research conducted in this area and identify any gaps in knowledge, a scoping review was conducted. The main research question was: What is the current knowledge in the literature about the effectiveness of ML models in identifying drug resistance? This includes identifying the types of algorithms and input data used in previous studies.

## 2. Materials and Methods

The literature was systematically reviewed to describe the usefulness of the newest contributions of computational ML methods in comprehensively diagnosing the *M. tuberculosis* complex. The review focused on the advantages of the technique for diagnosis, the most commonly used model, and the limitations of its implementation. The PRISMA methodology was implemented to review articles. The Covidence platform (https://app.covidence.org, (accessed on 1 March 2023)) was integrated to assist in elaborating reviews.

To find documents that might be relevant, we searched the following bibliographic databases (PubMed). The search results were exported to Covidence, and any duplicate entries were removed. Additionally, each article’s references were thoroughly searched to supplement the electronic database results. Search strategies were developed using Medical Subject Headings (MeSH) and text words related to ML and tuberculosis. Medline/Pubmed (National Center for Biotechnology Information, n.d.) and ScienceDirect (Elsevier, n.d.) databases were searched, and searches were limited to English. The reference lists of included studies or relevant reviews identified by the search were reviewed. Data were collected through a search taken from various PubMed publications, as well as WHO publications, including the words “whole genome sequencing” AND/OR “drug resistance prediction,” AND/OR “drug resistance learning,” AND/OR “tuberculosis incidence surveys,” AND/OR “genomic medicine,” AND/OR “clinical application of machine learning algorithms,” and analyzed against the criteria of direct relevance to the problem of drug resistance profiling using ML data on *M. tuberculosis* WGS.

### 2.1. Eligibility Criteria

Articles had to apply an ML model to genomic data to be included in the review. Articles from peer-reviewed journals were included if they were published between 2017 and 2022, written in English, and had at least a first-in-class drug analysis. The literature search results were uploaded into a collaborative file, and eligibility criteria and citation abstracts were attached to a database. Titles and abstracts produced by the search were independently screened against eligibility criteria. We thoroughly investigated all titles that appeared to meet the eligibility criteria or had uncertainties and obtained complete reports. Afterwards, we checked the full texts to ensure that they met the eligibility criteria. If needed, we contacted the study authors for extra information to clarify any questions about eligibility. Disputes were resolved through mediation. The reasons for excluding trials were recorded. Disagreements were settled through consensus. For each article, the following details were looked at: type of data, availability of data, number of observations, method of analysis, method of validation, and how the results matched up with direct methods (culture, PSD, rapid test). For each study included, the prediction performance metric was recorded concerning the directly measured value, which was taken as the confidence indicator for each method used.

To ensure consistency, we carefully reviewed and adjusted our selection and data extraction guidelines before starting the review process. Our reviewers worked in pairs to assess all publications related to our search, including their titles, abstracts, and full texts. Any disagreements were resolved through discussion and consensus with other reviewers.

### 2.2. Analysis of the Extracted Variables

In order to obtain the required variables, two reviewers collaborated and developed a data-charting form. They independently documented the data, shared their discoveries, and modified the form as necessary. A standardized data abstraction tool was utilized to record data from qualified studies. The Covidence platform gathered crucial information regarding the study’s characteristics and metrics pertaining to drug resistance.

We abstracted data on article characteristics (e.g., country of origin, funder), engagement characteristics, and contextual factors, determining 13 variables for the analysis (Table 1). For the variables “features” and “type of algorithm,” categories were defined in which the studies were grouped. The categories were defined in a two-step process: First, each study’s input data and the ML algorithm were evaluated in detail. From these detailed data, two reviewers defined meaningful and more general categories into which the studies were grouped. To ensure accuracy and relevance, we did not use standard textbook categorizations for the variables studied. We created our own categories that strike a balance between meaningful generalizations and detailed information. As a result, we identified ten categories for “type of algorithm” and five for “type of input data.” Keep in mind that a study could fall into more than one category depending on the algorithm type or input data used.

## 3. Results

According to the search criteria, seventy-one articles were found in the Medline/Pubmed and ScienceDirect databases. Of these, 57 were eliminated because they did not meet the eligibility criteria, and nine were eliminated because they needed to present a methodology or objectives for this research. In total, 24 articles were included for analysis (Figure 1). 

The study findings were divided into two categories: (1) diagnostic first-line drug resistance and (2) determination of second-line drug resistance. The criteria for grouping were based on the description of the ML method used, its association with a direct method, and its correspondence to clinical diagnosis (Table 1 and Table 2).

We identified 24 primary studies addressing research on ML models published between 2017 and 2022. There is a high degree of diversity between the training characteristics and the genomes’ number and origin. The most commonly used model is ANN [3,4,5,6,7,8,9,10,11,12]. However, in recent studies, ensemble learning models and ANNs have shown superior performance. The most used metric is accuracy. However, its predominance is 58.33%, showing no standardization between studies on which metric to use (Figure 2). Dimensionality reduction methods are prevalent in this data set due to their many characteristics. However, it is more prevalent (50%) to use non-statistical methods to reduce data, such as non-standardized criteria.

## 4. Discussion

After analyzing various models, we found that most of them use clustering for training and Random Forest for feature selection [4,5,6,7,8,9,10,13,14,15,16,17,18,19,20,21,22,23,24,25]. However, recent research shows that an increasing number of models are using neural networks [4,6,7,10,20,22], which have proven to be more effective. In fact, some studies have even implemented deep learning convolutional models [3,8,9,11], which show promising results. It’s worth noting that as the models become more complex, they become harder to standardize. This requires more abstract representations of biological features used for training. Clustering models use point mutations like SNPs, whereas deep learning models use representations based on scores [11] or feature ordering [9]. This may result in difficulty correlating relevant features during the classification process.

Our research revealed that there is a lack of focus on developing effective metrics to evaluate studies related to healthcare technology implementation. We also found a limited number of studies on the practical application of these technologies in clinical settings. Additionally, there is insufficient consideration given to the inclusion and empowerment of healthcare professionals in the education and use of these technologies. We did not come across any studies that address stakeholder relationships or the use of evaluative and iterative strategies to introduce and promote machine learning (ML) technologies in clinical practice. It was observed that the studies we reviewed invested more effort in improving analysis matrices than in performing standardized preprocessing that would enable comparison (Figure 2). 

The type of characteristics used to train the machine learning models varied for each study, from binary representation for the presence and absence of mutations in resistance genes or genes complete to physicochemical characteristics of the amino acids resulting from the base arrangement in each isolate [13]. Few studies compared the performance of the models for each characteristic used; this tells us about the need to generate a methodology that allows for a systematic comparison of the different types of characteristics used in training machine learning models. Systematically comparing each feature’s strengths and limitations and identifying which are most effective for specific machine-learning tasks will be easy for researchers. Additionally, it would allow researchers to better optimize the performance of their models by selecting the most appropriate characteristics.

Aytan-Aktug et al. (2020) [4] compared different feature representations, including binaries, scores, and combinations. They found that these representations slightly improved the prediction results, but the number of features differed significantly. For example, the amino acid representation had over 260,000 features, while the binary representation had only 6736. This stark contrast in the number of features raises the possibility of performing dimensionality reduction on the amino acid representation, which could further improve prediction results. Reducing the number of features could eliminate redundancies and noise, allowing for a more efficient and accurate representation of the data.

There is a wide range of approaches to analyzing mutations in these studies. Some focus on known resistance genes previously identified in the literature [4,8], while others explore the entire genome or specific types of single nucleotide polymorphisms (SNPs), deletions, or insertions [4,6,7,9,12,13,14,15,16,17,18,19,20,21]. Some studies look at mutations in a genome-wide context to discover new genes that may be related to resistance [22,23]. However, despite the various approaches taken, there needs to be more research that specifically examines the role of epistasis (the interaction between genes) in developing drug resistance, suggesting that there is a need for more studies that focus on understanding the complex genetic mechanisms that can contribute to the emergence of DR [10,11,24,26].

Most studies focus on model accuracy as their main performance metric. However, a predictor with a specificity and sensitivity of at least 95% [27] is generally required for clinical applications. However, this represents a significant challenge since most clinical data sets must be balanced between sensitivity and drug-resistant observations. Nevertheless, there is an imbalance, particularly noticeable for first-line TB drugs like isoniazid and rifampicin compared to other first- and second-line drugs. The model needs more examples in unbalanced sets to identify resistant cases, resulting in low sensitivity. While specificity can be high due to the model being presented with many sensitive cases, high accuracy in predicting sensitive cases does not necessarily imply good performance. This situation highlights the need for more robust modeling strategies to improve the specificity and sensitivity of predictive models for clinical implementation. Reporting these metrics in more standardized ways in reported models is also crucial.

Our research indicates the necessity for greater emphasis on creating measurable standards to assess studies and the limited attention given to implementing these studies in clinical settings. Additionally, technology inclusion and education for healthcare professionals are not being considered. We could not find any studies that address the importance of building relationships and using evaluative and iterative strategies while introducing and promoting machine learning technologies in clinical practice. Furthermore, the studies in our research tend to prioritize improving analysis matrices rather than standardized preprocessing for comparison purposes (refer to Figure 2).

TB continues to pose a significant global health burden, particularly with the emergence of drug-resistant strains. The COVID-19 pandemic has further underscored the urgent need to intensify efforts toward achieving the End TB strategy. However, diagnosing drug-resistant tuberculosis (DR-TB) has become increasingly challenging, despite the development of new diagnostic tools. The rising number of multidrug-resistant (MDR-TB) cases necessitates innovative approaches for accurate and timely detection of DR-TB.

In recent years, the integration of AI models has shown great promise in predicting DR-TB profiles with enhanced precision. These models leverage advanced algorithms to analyze complex genomic data, enabling the identification of key genetic mutations associated with drug resistance in *M. tuberculosis*. Although traditional diagnostic methods like Xpert MTB/XDR or Truenat, recommended by the WHO, have improved diagnostics, they may not detect all genome mutations associated with drug resistance. 

It is crucial to cover all aspects of the field to comprehensively understand the current landscape of AI models for predicting DR-TB profiles. Which includes exploring the diverse range of AI techniques employed, the datasets utilized, and the performance metrics used to evaluate their effectiveness. By examining experiments or studies of impact in sufficient depth, this review aims to analyze the strengths and limitations of existing approaches comprehensively. By delving into the experimental setups, data sources, and evaluation methodologies employed in these studies, valuable insights can be gained, allowing for a deeper understanding of the advancements and challenges in the field. Additionally, this review suggests new avenues for future research to address the current limitations and drive further progress in DR-TB prediction. By identifying gaps in knowledge and proposing novel research directions, researchers can pave the way for innovative solutions. These new avenues include the following:(a)Integrating multiple AI techniques or combining AI with other diagnostic modalities, such as imaging or transcriptomics, to improve prediction accuracy;(b)Developing more accurate and robust AI models that can handle complex and noisy data from different sources and settings;(c)Exploring the use of AI for predicting resistance to other drugs besides rifampicin, isoniazid, pyrazinamide, and fluoroquinolones;(d)Integrating AI with other technologies such as molecular diagnostics, biosensors, or nanotechnology for rapid and point-of-care detection of DR-TB;(e)Evaluating the cost-effectiveness, feasibility, and ethical implications of implementing AI for DR-TB diagnosis in low- and middle-income countries.

Moreover, efforts should be made to expand the diversity of training datasets, encompassing various geographic regions and genetic variants of *M. tuberculosis*, to enhance the generalizability of AI models. Furthermore, developing user-friendly bioinformatics analysis applications can simplify the interpretation of WGS data and facilitate widespread adoption of this technology.

Our scoping review has some limitations. The diversity of approaches to ML and the tendency to name the models with individual qualifiers cause some articles to not appear in the searches performed. To reduce this, we performed a manual search in the references of the articles and used tools such as research rabbit to be as exhaustive as possible. However, omissions may arise. In addition, many quick reviews contain confidential information and are not publicly available.

## 5. Conclusions

The small number of articles in the specific field of study and the heterogeneity of the results raise the need for further analysis to characterize the usefulness of ML methods as diagnostic aids. Little use of DL models is observed in the studies analyzed [4,6,8,9]. It was shown that it is possible to differentiate drug resistance or sensitivity using genomes; however, there is great diversity in the characteristics used to train the models. In the case of studies using models for mutation detection, it can be concluded that the prediction efficiency for each gene is different according to the model implemented. Therefore, integrating multiple models is more efficient than one. The most commonly used model is neural networks, which have quite variable accuracy and have shown high accuracy.

Nevertheless, only some articles were found that implement them, so further research is required for their characterization. This scoping review aimed to identify gaps in the literature that could guide a future systematic review. However, the lack of standardization in the approaches means that conducting a systematic review is neither appropriate nor necessary. We require high-quality research that considers the needs of the hospital setting to determine which techniques and models can be standardized and may be beneficial to this population and to guide clinicians on how to use these technologies.

In conclusion, this review aimed to cover all essential aspects of AI models for predicting DR-TB profiles, examine experiments or studies of impact in sufficient depth, and suggest new avenues for future research. By providing a comprehensive understanding of the current state of research, highlighting impactful studies, and proposing innovative directions, this review contributes to the ongoing efforts in combating DR-TB and ultimately achieving the goal of eliminating TB as a public health threat.

## Figures and Tables

**Figure 1 microorganisms-11-01872-f001:**
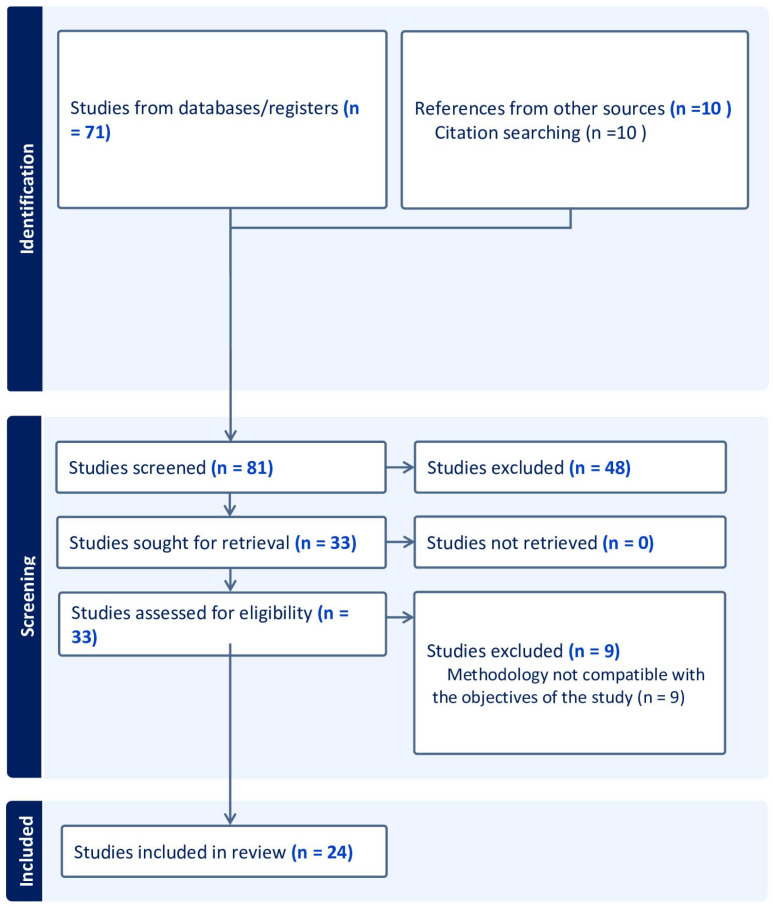
PRISMA flow chart of inclusion process. PRISMA: Preferred Reporting Items for Systematic Reviews and Meta-Analyses.

**Figure 2 microorganisms-11-01872-f002:**
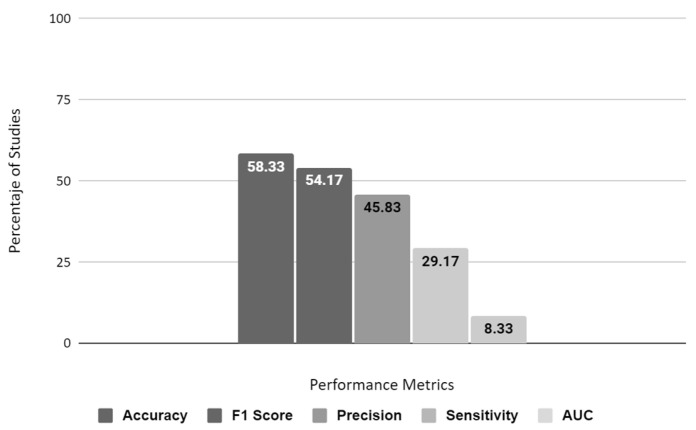
Metrics used in the state of the art.

**Table 1 microorganisms-11-01872-t001:** Data extracted from the studies.

Variable	Categories	Definition	Example
**Sample size**	<150, 150–1500, 1500–3600, 3600–8600, 8600–17,000, 17,000–32,700	Number of samples included	NA
**Publication year**	2017–2022	Year of the publication date of the article	2017–2022
**Country of study**	Countries	Country where the study was published	USA, Mexico, Brazil
**Input Data type**	Omic data	Second-generation sequencing platform output files	FastaQ files Illumina
	Sequence data	First-generation sequencing platform output files	FastaQ files Sanger
	Clinical data	Clinical records	SQL or any database
**Type of features**	Whole genome variants	All variants, including SNPs, deletions, and insertions	1296_ins_3_a_attc, fprA_564_del_2_acg_a
	Whole genome SNPs	Only variants previously classified as SNP	Known genetic positions registered in databases such as NC_000962.3: 1524 nt
	DNA variant	All variants of a resistance-related gene	*rpob, katG, embB*
	SNPs	All SNPs of a resistance-related gene	rpob_S450L, rpob_L430P, katG_R463L
	Catalog of mutation resistance	Genomic positions selected by a catalog of resistances published by WHO	Lys43Arg (aag/aGg)
**Number of features**	NA	Variables or attributes that are used to describe and quantify the input data that is used to train a machine-learning model	Binary or categorical representations of variants, complete sequence representations, and the number of patterns or relationships used for training.
**Origen of genomes**	Countries	Countries from which genomes were taken	
**Availability of data**	No available	There is no available data	
	Available	There are available data	The data or code used is provided through web pages or GitHub
**Type of algorithm**	Artificial Neural Network	Artificial intelligence method	Convolutional neural network, Recurrent neural network, multi-layer perceptron
	Bayesian Methods	A method of statistical inference	Naïve Bayes
	Clustering	The task involves organizing a set of objects into groups based on similarities between objects within the same group	k-means clustering, hierarchical clustering
	Decision tree	A graph that uses a branching method to illustrate every possible output for a specific input	Decision tree
	Discriminant analysis	A multivariate technique used to separate two or more groups of observations	Linear discriminant analysis
	Ensemble methods	Combines several base models	AdaBoost, Random Forest
	Instance-based learning	Family of techniques for classification and regression	k-nearest neighbor
	Logistic regression	Statistical analysis method to predict a binary outcome	Logistic regression
	Regression (Other)	Statistical processes for estimating the relationships between a dependent variable and one or more independent variables	Linear regression
	Kernel methods	This is a deep learning algorithm that uses supervised learning to classify or regress data groups	Support vector machine
	Other	Algorithms not classified into one of the categories above	Reinforcement learning, graphical models
**External Validation**	Yes	Performance of the algorithm tested on external data	Automated scoring of the genome with scoring by DST
	No	NA	NA
**Reduction method**	Not reduction	No dimensionality reduction methods were used.	NA
	Statistical	Statistical dimensionality reduction methods were used	PCA, RF, T-SNE
	Not statistical	Statistical dimensionality reduction methods were not used	Match catalogue
**Number of Drugs**	1–14	Number of drugs tested	NA
**Treatment line**	First line	OMS definition	RIF, INH, STR, EMB, PZA
	Second line	OMS definition	AMK, CAP, KAN, CIP, OFL, MOX, ETH, CYS, PAS
	Both lines	OMS definition	First and second line

**Table 2 microorganisms-11-01872-t002:** Frequency of type of algorithm implemented.

Type of Algorithm	%
Artificial Neural Network	28
Decision Tree	22
Clustering	13
Logistic Regression	6
Kernel Methods	6
Ensemble Methods	6
Bayesian Methods	3
Instance-Based Learning	3
Other	6

## Data Availability

The information provided in this study can be found within the article itself.

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
