# Peer review of "Machine Learning of the Whole Genome Sequence of Mycobacterium tuberculosis: A Scoping PRISMA-Based Review"

_microorganisms, 2023, doi:10.3390/microorganisms11081872_

Round 1

Reviewer 1 Report

Title: Machine learning of whole genome sequence of Mycobacterium tuberculosis: a scoping PRISMA-based review.

This manuscript deal with a hot topic in biomedical studies, such as machine learning. This topic most probably will lead the biomedical sciences to main changes on our understanding and interpretation of biological data. Authors made a very good revision of the literature that uses machine learning in the M. tuberculosis MDR field. They select several eligible criteria for homogeneity, aiming to help in data interpretation. Main features and relevance of a consensus procedure were highlighted. Finally, the strengths and limitations of current approaches were clearly identified.

The manuscript is very well written and well organized.

There are only few points that could need clarification/revision:

1.     Microbial names should be in italics. Please use “Mycobacterium tuberculosis” instead of “Mycobacterium tuberculosis”.

2.     Introduction, phrase in lines 62-63 has no clear meaning: “To help prompt clinical diagnoses…… from M. tuberculosis isolates”.

3.     Methods, information of Figure 1 is difficult to read.

Author Response

Comments and Suggestions for Authors

 This manuscript deal with a hot topic in biomedical studies, such as machine learning. This topic most probably will lead the biomedical sciences to main changes on our understanding and interpretation of biological data. Authors made a very good revision of the literature that uses machine learning in the M. tuberculosis MDR field. They select several eligible criteria for homogeneity, aiming to help in data interpretation. Main features and relevance of a consensus procedure were highlighted. Finally, the strengths and limitations of current approaches were clearly identified.

The manuscript is very well written and well organized.

There are only few points that could need clarification/revision:

  1. Microbial names should be in italics. Please use “Mycobacterium tuberculosis” instead of “Mycobacterium tuberculosis”.

Reply: The names of all microorganisms in the document are now written in italics.

  1. Introduction, phrase in lines 62-63 has no clear meaning: “To help prompt clinical diagnoses…… from M. tuberculosis isolates”.

Reply:

Last:

To help prompt clinical diagnoses based on drug-resistant profiles predicted directly from DNA-derived WGS data extracted from M. tuberculosis isolates.

New:

Machine learning models can be utilized to analyze the whole genome sequencing of M. tuberculosis strains, helping predict resistance profiles and reducing the time delay in starting appropriate treatment.

  1. Methods, information of Figure 1 is difficult to read.

Reply:
The image size has been increased.

Reviewer 2 Report

In this study the authors have provided comprehensive analysis of different AI programs available to detect drug resistant Mtb strains in tuberculosis patients. I think, they should mention also about development of drug resistant strains in lab settings and programs to predict this.

Author Response

Comments and Suggestions for Authors

In this study the authors have provided comprehensive analysis of different AI programs available to detect drug resistant Mtb strains in tuberculosis patients. I think, they should mention also about development of drug resistant strains in lab settings and programs to predict this.

Reply:

The subject of drug-resistant strains developing in lab settings and programs to predict them, as mentioned by the reviewer, is a fascinating topic that warrants further discussion. However, we need to gather additional scientific papers to enhance our responses. This should be addressed in a new paper revision.